# What mechanism design helps to realize the innovation function of maker-spaces: A qualitative comparative analysis based on fuzzy sets

**Jiancun Zheng**[1]*, **Lu Shi**[2], **Tianhong Jiang**[1]

**1** Economics and Management School, Wenzhou University of Technology, Whenzhou, China,
**2** Department of Business and Management, Wenzhou Vocational College of Science and Technology, Whenzhou, China

* zhengjiancun@139.com

**Data Availability Statement:** All relevant data are within the paper and its Supporting Information files.

## Abstract

Most of the existing studies on maker-space focus on internal subjects (such as makers) or external factors (such as policy support, ecological environment, and more). There has been relatively little discussion on the design of a series of mechanisms of maker-space. This paper theorizes the operating mechanism for platform services, resource gathering, network connections and endogenous cultural protection for the maker-space. It uses the method of fuzzy set qualitative comparative analysis (fsQCA) to analyze data from 63 maker-spaces in Zhejiang Province. The study proposes a reasonable mechanism design scheme for maker-space. The results show that the innovativeness of a maker-space is the result of the synergistic effect of various operating mechanisms. Among them, the platform service function, the channel for gathering resources, the formal linkages, and the culture for sharing achievements are indispensable support mechanisms for maker-spaces. Two effective ways to promote innovation in maker-space are outlined: first, preventing interventions from external resource providers; second, building an inclusive culture of trial and error.

## 1 Introduction

Maker-space refers to the place where office space or service support is available for various innovation and entrepreneurship activities [1]. In recent years, maker-spaces have flourished in China. They have become an important part of the national development strategy, which targets innovation. As an important carrier of "innovation and entrepreneurship", maker-space shoulders the mission of incubating start-up enterprises and improving the level of social innovation and development. However, currently, the development of maker-space is too dependent on government support. The spaces suffer from insufficient professional services, as well as other problems. The overall efficiency of their innovation is low [2]. Therefore, it is important to explore how innovation in maker-space can be encouraged.

**Funding:** The author(s) received no specific funding for this work.

**Competing interests:** The authors have declared that no competing interests exist.

As an innovation platform aimed at providing a stable innovation environment and incubating the growth of member enterprises, maker-space is essentially an innovation ecosystem [3]. Existing research on innovation ecosystem governance holds that different types of innovation ecosystems will have different degrees of "malfunction" [4,5]. Once the innovation ecosystem fails or lacks of effective operating mechanism, it will be difficult for its members to independently complete resource integration and value creation [6]. Wareham et al. (2014) and Jacobides et al. (2018) pointed out that innovation ecosystem governance system is the necessary condition for innovation platform to manage the relationship among its participants and realize value creation [7,8]. Therefore, the research on the development of maker-spaces should also enter a new stage of innovation governance from the early paradigm of science and technology policy. Thinking about the design of the micro-operation mechanism of maker-spaces will be the necessary work to promote the realization of the innovative function of maker-spaces [9].

Until now, research into the operating efficiency of maker-space has mainly focused on the following three aspects: the pioneering enterprises, the entrepreneurial platform, and the entrepreneurial ecology. In terms of the focus on pioneering enterprises, it is mainly concerned with the intrinsic motivation of makers [10], as well as the influence of a social network of start-up enterprises [11]. From the perspective of entrepreneurial platform, it focuses on the construction of service functions of the platform. These services include the supply of technology, financial support, training in entrepreneurship, the facilities for cooperation, and more [12,13]. The entrepreneurial ecology refers to how the external environment supports the construction and operation of maker-space. This environment includes government subsidies, external financing, external human capital (in the form of mentors), and more [14–16]. Generally speaking, the existing research mainly focuses on the influence of multi-dimensional subjects and multi-resource elements on the activities of maker-spaces, while the discussion on the internal mechanism of how to effectively integrate internal and external factors in maker-spaces is relatively scarce [17].

Based on the above considerations, this paper discusses how the operations of maker-space should be managed. Existing research and practical experience show that the operating mechanism is the result of "synergistic effect" of many factors. The effective operation of maker-space should be depend on the dynamic combination of various micro-mechanism elements. Therefore, this paper uses fuzzy set qualitative comparative analysis (fsQCA) to explore relevant combinatorial effects and interactive relations, assessing how they affect the innovativeness of maker-space. This method integrates multiple mechanisms into a common research framework to investigate the complex interactions that influence the array of operating mechanisms within maker-spaces.

## 2 Literature review

Previous studies have not systematically investigated the operation mechanisms of maker-spaces. Nevertheless, the scattered achievements involving the mechanism construction roughly include platform services, resource integration, network connection, cultural guarantee, user value creation, survival of the fittest, risk control [18,19], etc. According to the core functions of maker-spaces, platform service, resource integration and spiritual energy transmission [20], this paper focuses on analyzing the influence of four main mechanism elements of platform service, resource aggregation, network connection and cultural guarantee on the innovation performance of maker-spaces.

### 2.1 Platform service

There are three general forms that maker-spaces take: office spaces, value-added service platforms, and entrepreneurial ecosystem [20]. Among them, both value-added service platforms

and entrepreneurial ecosystems focus on constructing value-added services and offering entrepreneurial services. These services include organizing activities for exchange, arranging for entrepreneurship training, and helping with industrial and commercial taxes. For example, *Beijing Maker-space*, *Tencent Crowd Maker-space*, *Hangzhou Onion Capsule*, and other domestic maker-spaces pay attention to the establishment of platform services.

Currently, the majority of studies of these spaces have argued that platform services are the foundation of maker-space. They believe that they are the key factor that affects the rate of success of entrepreneurship and innovation [21,22]. However, certain spaces that focus on providing office space for makers have been very successful, namely *WeWork*, *SOHO 3Q*, *UrWork*, and more [23]. In contrast, the innovation performance of some maker-spaces which not only provide office services but also paid attention to platform service mechanism construction is not ideal [24]. Thus it can be seen that the construction of platform service function and the exertion of innovation function of maker-spaces are not simple binary relations of "0" and "1".

## 2.2 Resource gathering

Start-up enterprises are small and weak when they are new [25,26]. Therefore, their innovation and entrepreneurship are bound to encounter a bottleneck due to a lack of resources. A maker-space's ability to gather resources is very important to its level of innovation [27]. Most of the existing studies argue that maker-space should engage in resource gathering to make it easier for entrepreneurs to search for resources [10]. They should try and ensure that various resources flow into the space. This would help start-ups deal with their need for entrepreneurial knowledge, policy support, capital, and more [28]. The resource supply and innovation activities should be connected organically and integrated, especially the heterogeneous integration between strategic knowledge and technological resources, and the heterogeneous integration between resources and maker operation capabilities [18]. However, some scholars also argue that maker-space can still be innovative even if they have a limited ability to gather resources. This is because capital and knowledge can function as substitutes for maker-space [2]. Meanwhile, some studies have shown that it can be counterproductive for investors to try and help start-ups by intervening in their corporate governance [29]. Thus, resource aggregation mechanism may not play an entirely positive role in ensuring the innovativeness of maker-space.

## 2.3 Network connection

A maker-space is a consortium that gathers various kinds of stakeholders in resources such as innovation and entrepreneurship [30]. Its purpose is to reduce the cost of entrepreneurial activities by forming connections that provide guaranteed credit [31]. Therefore, for maker-space to be effective they must construct a solid network of connections. Studies have shown that a "collaborative governance organization" based on a contract among various subjects [32,33], or connections through project cooperation and task division [34], can be regarded as such a solid network. This is conducive to the exchange of information and knowledge, which helps start-ups to improve and become more innovative. However, some studies of social networking have also noted that not all network connections are good for business [35]. For start-ups, network embedding may bring "start-up network operating costs," which can negatively impact a company's innovation and performance. Excessive embedding may weaken an enterprise's own culture of innovation [36], or limit the scope of its opportunities for innovation [37]. Therefore, the network connection mechanism is not necessarily positively correlated with the innovativeness of a maker-space.

## 2.4 Cultural protection

Studies of entrepreneurial ecosystems have argued that a "soft environment" is crucial for ensuring that maker-spaces are innovative [38]. These environments emphasize sharing and create an ethos of tolerance and trial and error. First, maker-space are spaces for sharing values [39]. They should strengthen the construction of "sharing culture" on the basis of sharing physical space, helping to increase the exchange of new technologies, methods, and achievements across different companies or entrepreneurial teams. Second, maker-space should also encourage innovation in their spaces by creating an atmosphere that tolerates failure and recognizes the importance of trial and error [40]. Nevertheless, there have been relatively few empirical analyses of the impact of culture on the innovativeness of maker-space.

In response, this paper constructs a framework for the theoretical analysis of platform service, resource gathering, network connection, and cultural protection, on the innovativeness of maker-spaces (Fig 1). According to existing studies, it can be preliminarily determined that these four key elements influence the innovativeness of maker-spaces. However, how to interact and cooperate among these four mechanism elements, and whether there is substitutability between them, etc., are still issues that need to be discussed in depth. In this paper, therefore, fsQCA is used to explore various ways in which these four mechanisms could be implemented to promote innovation in maker-spaces.

## 3. Methodology

### 3.1 Research method

Qualitative comparative analysis (QCA) is a mixed-methods approach combining qualitative and quantitative analysis. The antecedent variables and result variables are conceptualized as sets using set theory. The membership values of each set of variables are obtained by calibrating the variables based on existing theories and practical experience. Then, the complex causal relationship between the combination of antecedent variables and result variables is analyzed [41]. This paper uses fsQCA, which is a form of QCA that uses fuzzy sets and truth tables. It differs from the traditional, binary logic of independent and dependent variables. It does not simply categorize results as 0 and 1 but deals with the fuzzy relationship between 0 and 1. Moreover, it can explain the relationship between the univariate and the result variable and also demonstrate how the combined interaction of multiple variables influenced the result variable [42]. It is therefore more consistent with the realities of social phenomena. It is capable of

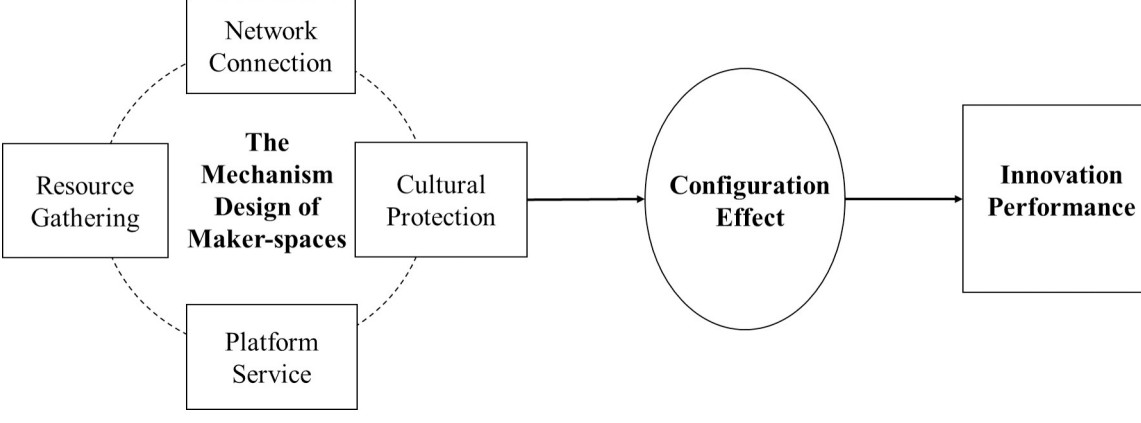

**Fig 1. Research model.**

accounting for the multiple and interdependent factors that go into creating synergy in complex management situations [43]. In this paper, therefore, fsQCA is used to reflect the various combinations of mechanisms that are conducive to creating innovation in maker-space.

At the same time, this paper also uses the newly emerging NCA to analyze the necessity of antecedent conditions, and to test whether a certain mechanism is the necessary condition for the creative function of mass maker-spaces, and the necessary level [44].

## 3.2 Sample selection and data sources

According to the *2019 China Torch Statistical Yearbook*, there are currently 3,867 maker-spaces in eastern China, accounting for 55.57% of the country's total. This reflects the current state of development of Chinese maker-spaces [45]. In view of the fact that Zhejiang is a representative region in eastern China and the availability of data in this province, this paper selects Zhejiang province as the focus of the study sample. 120 questionnaires were distributed to maker-space in Zhejiang province, of which 84 were recovered. Any incomplete questionnaires were eliminated, leaving 63 valid questionnaires. Statistical analysis of the sample data (Table 1) shows the number of jobs provided by maker-space, the scale of resident enterprises or teams, and the amount of intellectual property ownership. The sample level is consistent with the overall development level of maker-space in the eastern region, making it reflective of the overall situation of eastern China.

## 3.3 Measurement and calibration of variables

**3.3.1 Result variable.** As spaces for innovation and entrepreneurship, maker-spaces are tasked with promoting several different types of innovation: theoretical, institutional, service-based, cultural, and technological. Therefore, a maker-space's level of innovation is the primary indicator of its success. Studies vary in how they measure the innovativeness of maker-spaces, but the quantity of intellectual property produced in a particular space is a widely used measure [46,47]. Therefore, this paper sets the outcome variable (the innovativeness of a maker-space) as the quantity of intellectual property (QIP) obtained by enterprises or teams.

**3.3.2 Antecedent conditions.** This paper studies the index and interpretation of each antecedent variable as follows:

Platform service: the number of service functions (NSF) that maker-spaces provide is taken as the measurement index; that is, the number of different types of service that the maker-space provides to enterprises or teams alongside the provision of office space.

Resource gathering: this paper takes the number of channels for resource gathering (RGC) and heterogeneous resource integration (HRI) as the relevant measurement indicators. RGC is measured by the item number of the following three works: the construction of the declaration channels of normalized government policies, the implementation of entrepreneurship knowledge lectures and training, and the organization of project financing roadshows. HRI is judged

**Table 1. Sample description statistics versus overall data.**

| Project (number) | N | Sample average value | Overall average value |
|---|---|---|---|
| Number of jobs provided | 63 | 159.18 | 191.90 |
| Service enterprise/team size | 63 | 40.52 | 31.52 |
| Intellectual property number | 63 | 19.76 | 19.42 |

**Note**: Overall average value refers to the average of the overall situation of maker-spaces in eastern China. The data is taken from the *2019 China Torch Statistical Yearbook*.

according to whether investors provide additional guidance in addition to funding, such as management advice. These guidance work includes special guidance on enterprise development and governance provided by investors to the investee, or wide-ranging guidance on corporate governance by potential investors to many fund demanders.

Network connection: in this paper, the measurement index is whether there is a formal linkage (FL) established between each subject in the maker-space. An FL is any kind of business cooperation between different creators in the maker-space or any kind of common governance team.

Cultural protection: there are two types of cultural protection mechanisms. The first is a culture of achievement sharing (AS), and the second it a culture of trial and error (TE). AS considers whether there is a normalized exchange of achievements among the enterprises/teams in a particular maker-space. This exchange can involve the sharing of technical innovations, management experience, entrepreneurial experience, and more. TE considers whether or not the space provides subsidies for failed efforts to innovate, reductions for rent or service fees, introductions to venture capitalists, and other forms of help for creators struggling to innovate.

**3.3.3 Variable calibration.** Before running the fsQCA, it is necessary to calibrate each antecedent variable and result variable, assigning a membership score to each set of variables [41,48]. Due to the lack of explicit theoretical and empirical knowledge as a basis for calibration of the antecedent conditions and outcome variables, this paper uses objective quantile values to determine calibration anchors based on descriptive statistics of the cases. Draw on the experience of Fiss (2011) and Coduras et al. (2016), 0.95, 0.5 and 0.05 were taken as the critical values for full affiliation, crossover, and full disaffiliation, respectively [49,50]. The variables were calibrated using fsQCA3.0 software. The results are shown in Table 2.

## 4. Research analysis

### 4.1 Univariate necessity analysis

Before the configuration combination analysis of the antecedents, it is necessary to analyze the necessity of each antecedent to determine whether a certain condition always exists when the result occurs [41]. Using NCA method to analyze in this paper, we can not only judge whether the specific antecedent condition is the necessary condition of the result, but also get the bottleneck level of the necessary condition, that is, the level that the specific condition needs to reach under a certain result level. According to the advice of Dul (2016) and Dul et al.(2020), NCA can determine that a certain condition is a necessary condition when the effect d of the condition is not less than 0.1 and the result of Monte Carlo simulations of permutation test is significant [44,51].

**Table 2. Calibration of result and condition variables.**

| Variable | Goal set | Calibration anchors | | |
|---|---|---|---|---|
| | | **Full affiliation** | **Crossover** | **Full disaffiliation** |
| QIP | High innovation performance | 58 | 10 | 0 |
| NSF | Rich in service functions | 8 | 4 | 2 |
| RGC | There are many collection channels for resource gathering | 3 | 2 | 0 |
| HRI | Strong ability of heterogeneous resource integration | 2 | 1 | 0 |
| FL | Have formal connections | 1 | / | 0 |
| AS | Have a culture of sharing achievement | 1 | / | 0 |
| TE | Have a culture of trial and error | 1 | / | 0 |

**Table 3. Analysis results of necessary conditions of NCA.**

| Conditions[a] | Method | Precision | Ceiling zone | Scope | Effect size (d)[b] | P-value [c] |
|---|---|---|---|---|---|---|
| NSF | CR | 100% | 0.2 | 0.83 | 0.241 | 0.003 |
| RGC | CE | 100% | 0.583 | 0.83 | 0.704 | 0 |
| HRI | CE | 100% | 0.04 | 0.83 | 0.049 | 0 |
| FL | CE | 100% | 0.47 | 0.92 | 0.051 | 0.337 |
| AS | CE | 100% | 0.04 | 0.92 | 0.043 | 0.068 |
| TE | CE | 100% | 0.3 | 0.92 | 0.226 | 0.142 |

**a**. post-calibration fuzzy set membership value.

**b**. $0.0 \leq d < 0.1$ indicates low level, $0.1 \leq d < 0.3$ indicates medium level.

**c**. permutation test in NCA analysis (Re-pumping times = 10000).

According to the requirements of NCA analysis, when both X and Y are continuous variables or discrete variables of level 5 or above, the upper bound function is generated by ceiling regression (CR), otherwise, the ceiling envelopment analysis (CE) is used to generate the function. Among the variables involved in this study, the result variable QIP and the antecedent variable NSF are continuous variables, and the remaining variables are binary variables or discrete variables below level 5. The analysis results are shown in Tables 3 and 4.

According to the results in Table 3, the effects of NSF and RGC are both greater than 0.1, and the Monte Carlo simulations of permutation test shows that the effects are significant (P values are 0.003 and 0 respectively), which indicates that NSF and RGC are the necessary conditions for maker-spaces to achieve high innovation performance. For HRI, although the Monte Carlo simulation permutation test shows that the effect quantity is significant, the value of the effect quantity D is less than 0.1, which does not meet the necessary criteria. Other conditions are all not necessary for the high innovation performance of maker-spaces.

Table 4 shows the bottleneck level of each condition under NCA method. If we want to achieve 50% of maker-spaces' innovation performance level, we need 2% level of the NSF, 50.1% level of the RGC and 2.1% level of the FL, while there is no bottleneck level for other conditions at this time.

At the same time, this paper also adopts the necessary condition test of QCA. Results As shown in Table 5, the consistencies of the NSF and RGC both are greater than 0.9, which

**Table 4. Analysis results of bottleneck level of NCA.**

| QIP | NSF | RGC | HRI | FL | AS | TE |
|---|---|---|---|---|---|---|
| 0 | NN | NN | NN | NN | NN | NN |
| 10 | NN | 9.8 | NN | NN | NN | NN |
| 20 | NN | 19.9 | NN | NN | NN | NN |
| 30 | NN | 30 | NN | NN | NN | NN |
| 40 | NN | 40.1 | NN | NN | NN | NN |
| 50 | 2 | 50.1 | NN | 2.1 | NN | NN |
| 60 | 20.5 | 60.2 | NN | 4.7 | NN | NN |
| 70 | 39 | 70.3 | NN | 7.3 | NN | 8 |
| 80 | 57.5 | 80.4 | NN | 9.9 | NN | 18.7 |
| 90 | 76 | 90.5 | NN | 12.5 | NN | 29.4 |
| 100 | 94.4 | NA | 1.8 | 15.1 | 1.1 | 40.1 |

**Note**: **a**. CR method. NN = unnecessary.

Table 5. Necessary condition analysis for univariate antecedent variable.

| Antecedent conditions | Innovativeness of maker-space (QIP) | | Antecedent conditions | Innovativeness of maker-space (QIP) | |
|---|---|---|---|---|---|
| | consistency | coverage | | consistency | coverage |
| NSF | 0.93189 | 0.53976 | FL | 0.79459 | 0.81667 |
| ~NSF | 0.24541 | 0.45129 | ~FL | 0.20541 | 0.15833 |
| RGC | 0.99676 | 0.73760 | AS | 0.82811 | 0.47875 |
| ~RGC | 0.25405 | 0.27647 | ~AS | 0.17189 | 0.31800 |
| HRI | 0.73081 | 0.64381 | TE | 0.77297 | 0.65000 |
| ~HRI | 0.44757 | 0.39429 | ~TE | 0.22703 | 0.21000 |

Data source: fsQCA3.0 software.

indicates that the platform service mechanism and resource gathering channel are the necessary conditions for maker-spaces to produce high innovation performance, and have been explained in many cases (54% and 74% respectively). The QCA results are consistent with NCA results.

## 4.2 Configuration analysis

In this paper, fsQCA3.0 software was used to import each antecedent variable. The case threshold was set as 1, and the consistency threshold is set as 0.8, the PRI threshold is set as 0.7. The complex solution, intermediate solution, and concise solution were obtained. The antecedent variable configuration combination results for innovation in maker-spaces are shown in Table 6. In this table, ⊚ represents the core condition, that is, the condition contained in both the concise solution and the intermediate solution. ○ represents the auxiliary condition, that is, the antecedent condition contained only in the intermediate solution. ● denotes the absence of any auxiliary conditions. A blank indicates that the existence of the condition did not affect the result.

As can be seen from Table 6, the results provide two paths to promote innovation in maker-space. The overall consistency is 0.8920, and the overall coverage rate is 0.6605. In other words, the interpretation degree of the two operation mechanism schemes for the implementation of innovation in maker-space is 89.20%, covering 66.05% of all cases. From the

Table 6. The configuration of the function realization mechanism of maker-space.

| Antecedent conditions | | configuration 1 | configuration 2 |
|---|---|---|---|
| Platform service mechanism | Number of service functions (NSF) | ○ | ○ |
| Resource gathering mechanism | Resource gathering channel (RGC) | ⊚ | ⊚ |
| | Heterogeneous resource integration (HRI) | ● | |
| Network connection mechanism | Formal linkages (FL) | ⊚ | ⊚ |
| Cultural protection mechanism | Achievement sharing culture (AS) | ○ | ○ |
| | Trial-and-error culture (TE) | | ⊚ |
| Consistency | | 0.8431 | 0.9376 |
| Raw coverage | | 0.5524 | 0.4281 |
| Unique coverage | | 0.2324 | 0.1081 |
| Solution coverage | | 0.8920 | |
| Solution consistency | | 0.6605 | |

Data source: fsQCA3.0 software.

**Table 7. The configuration of consistency level adjusted from 0.8 to 0.81.**

| Antecedent conditions | | configuration 3 |
|---|---|---|
| Platform service mechanism | Number of service functions (NSF) | ○ |
| Resource gathering mechanism | Resource gathering channel (RGC) | ○ |
| | Heterogeneous resource integration (HRI) | |
| Network connection mechanism | Formal linkages (FL) | ○ |
| Cultural protection mechanism | Achievement sharing culture (AS) | ○ |
| | Trial-and-error culture (TE) | ⊚ |
| Consistency | | 0.9376 |
| Raw coverage | | 0.5524 |
| Unique coverage | | 0.5524 |
| Solution coverage | | 0.5524 |
| Solution consistency | | 0.9376 |

Data source: fsQCA3.0 software.

perspective of the two specific configurations, the consistency of configuration 1 is 0.8431, and the consistency of configuration 2 is 0.9376. The raw coverage is 0.5524 and 0.4281 respectively. This shows that the two paths found in this study could help nearly half of the cases to achieve innovation. Both paths would also be highly reliable.

In configuration 1 (NSF*RGC*FL*AS*~HRI), resource gathering channels and formal linkages play central roles. This is assisted by platform service functions, achievement sharing culture, and the absence of heterogeneous integration. In configuration 2 (NSF*RGC*FL*AS*TE), resource gathering channels and formal linkages still play a core role, while platform service functions, achievement sharing culture, and trial-and-error culture play auxiliary roles. In terms of a horizontal comparison, it is clear that resource gathering channels, formal linkages, platform service functions, and achievement sharing culture play the same role in both paths. Meanwhile, the absence of heterogeneous integration and trial-and-error culture are auxiliary conditions that have a certain substitutability.

## 4.3 Robustness test

To test the robustness of the results, the consistency level was adjusted from 0.8 to 0.81 [52]. The result turned out to be configuration 3 (NSF*RGC*FL*AS*TE) as shown in Table 7, and its overall coverage and consistency have only slightly changed, and configuration 3 and configuration 2 are consistent in antecedent condition types and number combinations. Although there are some changes in core and auxiliary roles, it has not caused substantial changes to the mechanism design of maker-space. According to Schneider and Wagemann(2012) [48], the research conclusion can be considered to be relatively robust.

## 5 Discussion

The fsQCA shows that there are four indispensable mechanism design requirements for promoting innovation in maker-space. Also, there are two paths of operation mechanism design that can promote innovation in maker-spaces. This suggests that there are multiple possible effective designs for maker-space. Based on the conditions and the logic behind these two paths, this paper refers to them as the shielded external intervention mode and the internal flexible culture mode.

## 5.1 Necessary mechanism elements

Based on the results of the fsQCA, there are four identical conditions in the two feasible operation mechanism designs. They are indispensable to promoting innovation in maker-space and are part of the necessary mechanism design. We call it "the necessary mechanism elements". They are platform service functions, resource gathering channels, formal linkages, and the culture of sharing achievements. Among them, resource gathering channels and formal linkages are the core conditions, whereas platform service functions and achievement-sharing culture are auxiliary conditions. To promote innovation in maker-space it is necessary to optimize the perfect platform service function, introduce a wide range of resource gathering channels, encourage formal linkages, and create an atmosphere of sharing achievements.

## 5.2 Shielded external intervention mode

The shielded external intervention mode (NSF*RGC*FL*AS*~HRI) indicates that the maker-spaces need to be shielded from intervention by external resource providers. This approach is based on the necessary mechanism elements with heterogeneous resource integration absent. This finding is supported by other existing studies. For example, researchers have argued that some low-reputation investors may steal or take over the interests of small and medium-sized enterprises, which can harm corporate innovation [53,54]. Also, the participation of investors in corporate governance may lead to "excessive supervision," which can also stifle innovation [55]. Most of the parties working in maker-spaces are start-ups or small, medium, or micro-enterprises. Their managers are the main strategic planners who implement their company's innovative behaviours. When a resource provider participates in the integration of heterogeneous resources, it can interfere with or even destroy the integrity of the original manager's plan for innovation. This mode emphasizes the importance of shielding the enterprises from intervention from external resource providers, thereby safeguarding the ability of those enterprises to innovate.

## 5.3 Internal flexible culture mode

The internal flexible culture mode (NSF*RGC*FL*AS*TE) indicates that the combined effect of platform service functions, resource gathering channels, formal linkages, achievement-sharing culture, and trial-and-error culture can promote innovation in maker-space. Based on the necessary mechanism elements, this mode emphasizes the role of a culture that tolerates trial and error. Maker-spaces are essentially trial-and-error laboratories. Compared with incubators, there are almost no thresholds for entering the field [22]. For this reason, the rate of failure within maker-space is relatively high. The levels of innovation in a maker-space are often very high when the space is first established, but as many projects fail, those levels gradually drop off. Tolerating a culture of trial and error in can help to produce sustainable levels of innovation in maker-space [40]. If a maker-space can build a flexible internal culture on the basis of necessary mechanism design, it can maintain the innovation vitality to the maximum extent and improve the innovation performance.

## 6 Conclusions and implications

The main purpose of this study was to investigate the rational mechanism design behind the complex operation activities of maker-spaces as entrepreneurial ecosystems. Through theoretical analysis and combing of existing research, according to the core functions of maker-spaces, focusing on the synergy effect of four mechanism elements, namely platform service, resource gathering, network connection and cultural protection, in the process of high innovation

performance of maker-spaces, this paper analyzes 63 maker-spaces in Zhejiang Province of China by using the mixed method of fsQCA and NCA, and draws the following conclusions:

The innovativeness of maker-spaces is the result of the synergistic effect of various operating mechanisms. There are several ways to create that synergy. One single mechanism is not sufficient on its own to ensure that a maker-space is effective and innovative. This study finds that platform service functions, resource gathering channels, formal linkages, and a culture of sharing achievements are indispensable to maker-spaces. Based on the design of the four necessary mechanisms, shielding the intervention behaviours of external resource providers or building an inclusive culture of trial and error are both effective paths to promote the innovation function of maker-spaces.

Shielding companies or entrepreneurial teams against external intervention mode is a feasible mechanism for ensuring innovation in maker-spaces. In previous studies, external resource providers, especially venture capitalists, have sometimes been thought to have considerable operational knowledge and useful management experience. These studies have therefore argued that involving these external resource providers in the corporate governance of companies would have a positive impact [18,56]. However, the results of this study indicate that for the start-ups or small, medium, or micro-enterprises in the maker-space, the participation of resource providers through heterogeneous resource integration can harm the innovativeness of the smaller enterprises. Maker-space should therefore protect companies or entrepreneurial teams against this.

An internal flexible culture is also an effective mechanism for promoting innovation. Specifically, the creation of a flexible internal culture that is tolerant of a trial-and-error approach can help to ensure that maker-spaces remain innovative. This conclusion has been confirmed by existing studies [38,40].

## 7 Limitations and future research

This study has certain limitations. First, the existing research on the internal operation of maker-spaces is scarce, and has not formed a complete theoretical system on the internal operational mechanism. The four operational mechanism elements that this paper focuses on are refined based on the sorting out of previous scarce and scattered literature. There may be some elements that have been omitted in this study because previous studies haven't paid attention to them, and some of these factors may have a noticeable impact on maker-space's innovative performance. The follow-up work needs to do more detailed and in-depth research in this respect. Second, the data collected and used in this study are cross-sectional data at specific time points, which can be used to discuss the reasonable operation mechanism configurations under the current innovation performance. However, it is difficult to analyze and judge the long-term impact of specific mechanism design on the innovation performance of maker-space from these data. Therefore, future research can consider collecting longitudinal data and combining other methods to bring the time dimension into research. In addition, the research samples selected in this study are concentrated in specific areas, and the differences of regional culture and regional policies may affect the mechanism design of maker-space. Future research can consider expanding the scope of sample selection, and cross-regional samples may bring something new.

## Supporting information

**S1 Data.**
(XLSX)

## Author Contributions

**Data curation:** Lu Shi, Tianhong Jiang.

**Formal analysis:** Lu Shi, Tianhong Jiang.

**Investigation:** Jiancun Zheng, Lu Shi, Tianhong Jiang.

**Methodology:** Lu Shi, Tianhong Jiang.

**Project administration:** Jiancun Zheng.

**Resources:** Tianhong Jiang.

**Software:** Lu Shi.

**Supervision:** Jiancun Zheng.

**Validation:** Lu Shi.

**Visualization:** Lu Shi.

**Writing – original draft:** Jiancun Zheng, Lu Shi.

**Writing – review & editing:** Jiancun Zheng, Lu Shi.

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
