## [Decision Letter · Decision Letter 0]

19 Jul 2022

PONE-D-21-37616What mechanism design helps to realize the innovation function of maker-spaces: A qualitative comparative analysis based on fuzzy setsPLOS ONE

Dear Dr. Zheng,

Thank you for submitting your manuscript to PLOS ONE. After careful consideration, we feel that it has merit but does not fully meet PLOS ONE’s publication criteria as it currently stands. Therefore, we invite you to submit a revised version of the manuscript that addresses the points raised during the review process.

We look forward to receiving your revised manuscript.

Kind regards,

Hao Xue

Academic Editor

PLOS ONE

Journal Requirements:

Reviewers' comments:

Reviewer's Responses to Questions

**Comments to the Author**

1. Is the manuscript technically sound, and do the data support the conclusions?

Reviewer #1: Yes

Reviewer #2: Yes

2. Has the statistical analysis been performed appropriately and rigorously? 

Reviewer #1: Yes

Reviewer #2: Yes

3. Have the authors made all data underlying the findings in their manuscript fully available?

Reviewer #1: Yes

Reviewer #2: Yes

4. Is the manuscript presented in an intelligible fashion and written in standard English?

Reviewer #1: Yes

Reviewer #2: Yes

5. Review Comments to the Author

Reviewer #1: The manuscript is well-written, methodologically sound, and has some very good aspects to consider. The authors have provided a detailed description of the methods used, and results are well-reported.

Reviewer #2: Review comments PONE-D-21-37616

Thank you for giving me the opportunity to read this exciting study! This manuscript titled “What mechanism design helps to realize the innovation function of maker-spaces: A qualitative comparative analysis based on fuzzy sets” uses the method of fuzzy set qualitative comparative analysis (fsQCA) to analyze data from 63 maker-spaces in Zhejiang Province and to study proposes a reasonable mechanism design scheme for maker-space. Given the unique research design, the rareness of the maker-spaces data, and robust results, there is no doubt that it is an important paper and an interesting story to read. My comments are as follows:

Major comments:

1. This manuscript has a rigorous for the methodology and research analysis section. I still have some concern on part of it. In line 315-316, and line 318-321, the authors mentioned that they use the raw continuous data, binary data, and level data. Since the sample size is small, it would be great if the author can conduct some robustness analysis to make the results more robust. For example, for the continuous result data, taking logarithm would make the estimate more tolerate to outliers.

2. In the robustness test section, it would be great if the author could add a table here or in the appendix to show the robustness check results.

Minor comments:

3. This manuscript has a rigorous contribution to the literature by conducting surveys with 63 valid questionnaires. If allowed, a larger sample size would contribute to a more robustness results.

4. It’s nice that the author also discussed the future research and limitations at the end of the paper. It would be helpful if the limitation is discussed in more details.

5. It might be helpful for readers to clearly understand the results if the figure for the results in the main tables is added.

6. PLOS authors have the option to publish the peer review history of their article (what does this mean?). If published, this will include your full peer review and any attached files.

Reviewer #1: **Yes: **Denny John

Reviewer #2: No

---

## [Author Response · Author response to Decision Letter 0]

30 Jul 2022

Dear editor and reviewer(s),

We thank you for your generous comments on the manuscript, and have edited the manuscript to address your concerns.

According to the review comments, the revised content has been marked yellow in the new manuscript. The following is feedback on the review comments. 

Major comment 1:

This manuscript has a rigorous for the methodology and research analysis section. I still have some concern on part of it. In line 315-316, and line 318-321, the authors mentioned that they use the raw continuous data, binary data, and level data. Since the sample size is small, it would be great if the author can conduct some robustness analysis to make the results more robust. For example, for the continuous result data, taking logarithm would make the estimate more tolerate to outliers.

As reviewers can see, this paper does involve different types of data, including continuous data, binary data and level data. As for the concerns about the possible data robustness, in fact, the data analysis methods we have adopted have fully considered the characteristics of different types of data and avoided the problem of outliers. In the application of NCA(Necessary Condition Analysis), according to the application requirements of this method, when both X and Y are continuous variables or discrete variables of level 5 or above, the upper bound function is generated by ceiling regression (CR), otherwise, the ceiling envelopment analysis (CE) is used to generate the function. We can see this kind of treatment in Table 3 of the manuscript. In the application of QCA, according to the application requirements of QCA technology, all types of data are calibrated, so that each data becomes the membership score of the corresponding case on a specific variable set, thus eliminating the differences of data types and the robustness concerns brought by data types.

In addition, 63 samples were collected in this paper, which is small compared with the traditional statistical analysis method, but reasonable compared with the QCA analysis method adopted in this paper. Traditional statistical analysis is based on large samples, maximum random process and relatively few variables. The QCA method is a method that integrates qualitative analysis and quantitative analysis. This method is case-oriented, analyzing and handling a limited number of complex cases through configuration, and it is a suitable method to handle "small sample" research (Rihoux and Ragin, 2009). In this paper, 63 samples were analyzed by QCA method, which reached the level of "medium samples" in the application of QCA technology. Therefore, the research conclusions obtained should be reliable.

Major comment 2:

In the robustness test section, it would be great if the author could add a table here or in the appendix to show the robustness check results.

Agreed. According to the suggestion of the reviewer, we added a data table of robustness test to the manuscript (Table 7). Please refer to the revised manuscript.

Minor comment 3:

This manuscript has a rigorous contribution to the literature by conducting surveys with 63 valid questionnaires. If allowed, a larger sample size would contribute to a more robustness results.

We very much agree with the reviewer's suggestion of expanding the sample size. Enlarging the sample size as much as possible is more conducive to research work. In this study, the team conducted three rounds of questionnaire to get enough sample cases. However, for various reasons, we regret that only 63 valid samples were obtained. We are willing to do our best to expand the sample size in follow-up studies, such as cross-regional study.

Minor comment 4:

It’s nice that the author also discussed the future research and limitations at the end of the paper. It would be helpful if the limitation is discussed in more details.

According to the suggestion of the reviewer, we tried our best to explain "limitations and future research" in more detail. Please refer to the revised manuscript for details.

Minor comment 5:

It might be helpful for readers to clearly understand the results if the figure for the results in the main tables is added.

QCA technical analysis mainly focuses on the analysis of whether there are reasonable configurations of various antecedents under an interesting result, and its analysis process data is relatively few (Rihoux and Ragin, 2009). In this paper, fsQCA 3.0 software is used to analyze the mechanism configuration that is conducive to the innovation performance of maker-space. The results given by the software are as follows: " fsNSF * fsRGC * ~ fsHRI * FL * AS " and the corresponding consistency and coverage data. To facilitate readers' understanding, we adopted the common methods in previous studies (Greckhamer, 2016, Delmas and Pekovic, 2018), and presented the main table of results in the form of tables, which are composed of various symbols and corresponding consistency and coverage (Table 6). All digital information generated by the analysis has been included. Therefore, we are sorry that we can't add any more data to the main table. In the follow-up research, we are willing to combine other research methods to increase the process data according to the reviewer's suggestion, so as to make our research more perfect.

Yours sincerely,

Jiancun Zheng

---

## [Editor Report · Decision Letter 1]

26 Aug 2022

What mechanism design helps to realize the innovation function of maker-spaces: A qualitative comparative analysis based on fuzzy sets

PONE-D-21-37616R1

Dear Dr. Zheng,

We’re pleased to inform you that your manuscript has been judged scientifically suitable for publication and will be formally accepted for publication once it meets all outstanding technical requirements.

Kind regards,

Hao Xue

Academic Editor

PLOS ONE

---

## [Editor Report · Acceptance letter]

30 Aug 2022

PONE-D-21-37616R1 

What mechanism design helps to realize the innovation function of maker-spaces: A qualitative comparative analysis based on fuzzy sets 

Dear Dr. Zheng:

I'm pleased to inform you that your manuscript has been deemed suitable for publication in PLOS ONE. Congratulations! Your manuscript is now with our production department. 

Kind regards, 

on behalf of

Dr. Hao Xue 

Academic Editor

PLOS ONE